# Effects of brief mindfulness intervention on mental fatigue and recovery in basketball tactical performance

**Shudian Cao**[1], **Jia Liu**[2,3]*, **Soh Kim Geok**[4], **He Sun**[5], **Xiaopeng Wang**[4]

**1** School of Physical Education, Xihua University, Chengdu, China, **2** Department of Physical Education, Yuncheng University, Yuncheng, China, **3** College of Physical Education, Changwon National University, Changwon, South Korea, **4** Faculty of Educational Studies, University Putra Malaysia, Selangor, Malaysia, **5** School of Physical Education, Henan University, Kaifeng, China

\* liujia1986yuncheng@163.com

**Data Availability Statement:** All relevant data are within the manuscript and its Supporting Information files.

**Funding:** Talent Introduction Project of Xihua University, No: W2420096.

## Abstract

### Objective

The detrimental effects of mental fatigue (MF) have been established in sports, such as soccer, volleyball, and basketball. Mindfulness interventions are considered a promising method to help players counteract MF, but whether it could improve basketball tactical performance after MF in competition is not clear. This study aims to investigate the effect of brief mindfulness intervention on basketball tactical performance under MF.

### Method

This study employed a cluster randomized controlled trial (cRCT) design. It involved 54 male basketball players aged 18 to 24 from three universities. The participants were randomly assigned to one of three groups: control group (CG), mental fatigue group (MFG), and mental fatigue-mindfulness group (MF-MG). Players in the MFG and MF-MG underwent a 30-minute Stroop task to induce MF. Subsequently, players in the MF-MG engaged in a 30-minute audio mindfulness intervention. Basketball tactical performance was assessed in the small side games (SSG).

### Results

There were no significant differences in total tactics observed across groups and over time. However, when examining specific tactical sub-variables, significant differences were found in ball reversal, dribble penetration into the key area, and off-ball screen between the CG and MFG in the post-test. Furthermore, significant differences were noted in ball reversal, dribble penetration into the key area, on-ball screen, and off-ball screen between the MFG and MF-MG in the post-test.

### Conclusion

The basketball tactical performances, particularly in areas such as ball reversal, dribble penetration into the key area, on-ball screen, and off-ball screen, were negatively impacted by

**Competing interests:** The authors have declared that no competing interests exist.

MF. Notably, the brief mindfulness intervention effectively restored these performance aspects. This suggests that coaches and trainers should place increased emphasis on players' mental well-being and consider incorporating brief mindfulness interventions into their training programs. More studies that investigate mindfulness intervention on the comprehensive aspects of basketball performance should be focused on in the future.

## 1. Introduction

"Fatigue" is a term used to describe a state of tiredness or exhaustion. Due to its wide usage and varying interpretations, fatigue encompasses many definitions and concepts. In general, fatigue often follows prolonged physical or mental activity. Both physical and mental domains of fatigue have significant effects on human performance, resembling a medical condition [1, 2].

In recent decades, mental fatigue (MF), defined as a psychobiological state induced during extended, demanding cognitive activities leading to a subjective sense of tiredness, reduced cognitive capacity, and altered brain activation, has garnered substantial attention [3]. MF impairs sports-related performance, affecting motor skills (accuracy and speed) [4, 5], endurance (such as cycling and the yoyo test) [6, 7], maximal force production [8, 9], and decision-making (resulting in errors) [10–12]. On the other hand, MF has negative effects on psycho-physiological factors. For instance, MF results in decreased concentration, increased distractibility, and a reduction in the ability to perform tasks requiring sustained attention. It also impairs cognitive functions, making it harder for individuals to focus, process information efficiently, and maintain alertness [13, 14]. On the other hand, Motivation is a key factor in determining the effects of MF. It can drive individuals to engage in the activities. An individual could not maintain their performance level if they were no longer motivated [3, 15], but extrinsic motivation, like offering a monetary reward, can counteract this negative effect of MF [16]. However, a study showed that the MF did not impact success motivation and intrinsic motivation but increased the perception of effort to impair the sports performance [17], which was consistent with the psychobiological model of exercise performance based on motivational intensity theory [18, 19]. Only cognitive activity can affect the motivation related to the tasks, and a study also showed that the physical fatigue from subsequent physical tasks reduced the willingness to exert effort but not mental fatigue [17, 20].

In basketball, characterized by high psychological and cognitive demands during intense competitions, players need to exert self-control to manage anxiety and focus. Overexertion of self-control and the constant state of alertness demanded by anxiety may lead to ego depletion or MF [21, 22]. The negative effect of MF on basketball performance has been proved [23]. Filipas et al. (2021) and Shaabani et al. (2020) showed that MF induced a reduction in amateur basketball free throw performance [24, 25]. Bahrami et al. (2020) indicated that MF could hurt basketball athletes' three-point shoot [26]. Hepler & Kovacs, (2017) illustrated that MF impaired the decision-making speed of basketball players [27]. Regarding the impact of MF on tactical performance, Coutinho et al. (2017) demonstrated that MF impairs players' ability to utilize environmental information and positioning in soccer [28]. Silva et al. (2022) reported that pre-induced MF reduces the number of tactical actions (with a significant effect on the total of offensive actions), but the quality is enhanced (with a medium effect on the efficiency of all offensive actions) because players may have been more selective based on the theoretical model that mental fatigue can increase the subjective perception of effort [29]. However, the effects of MF on basketball tactical performance remain unclear. Reaction time is crucial for

tactical success. Some studies indicated that MF slows simple and complex reaction times, affecting the ability to make timely decisions and execute accurate responses [30, 31]. In addition, MF could reduce attentional focus, leading to delayed reactions and increased error rates [13].

Mindfulness meditation has proven to be an effective method for preventing MF. Mindfulness involves intentionally directing one's attention to the present moment without judgment [32]. The positive effect of mindfulness on MF has been proved [33]. One study showed that the mindfulness intervention effectively attenuated the MF caused by competition in volleyball athletes [34]. Zhu et al. (2020) indicated that mindfulness-based intervention recovered the MF in the second half of soccer game [35]. Axelsen et al. (2020) indicated that mindfulness intervention improved the mind-wandering under MF [36]. On the other hand, by fostering greater awareness of the present moment, players can experience more enjoyment and satisfaction from their training and competition, which enhances their intrinsic motivation and decreases perceived exertion to engage in and persist with their sport [37–39]. In addition, mindfulness practices, such as meditation, can enhance an individual's ability to regulate emotions and respond to stressors more calmly and effectively, thereby reducing anxiety. By reducing anxiety, mindfulness can indirectly alleviate some of the mental fatigue associated with chronic stress. Simultaneously, by addressing mental fatigue, mindfulness can improve cognitive function and emotional regulation, which are essential for managing anxiety [40]. However, implementing mindfulness interventions can sometimes pose challenges. Interventions like Mindfulness-Acceptance and Commitment (MAC) and the Mindfulness Sport Performance Enhancement Program (MSPE) often require several weeks to complete and are typically administered in specialized settings by highly trained therapists [41, 42]. To address these feasibility concerns, brief mindfulness interventions with shorter durations and lower intensity have been developed.

Although it has been proved that brief mindfulness intervention can mitigate the negative effects of MF on basketball free-throw performance [25], the effect of brief mindfulness intervention on basketball tactical performance under MF remains unclear. The present study aims to investigate whether a brief mindfulness training can counteract the detrimental effects of MF on basketball tactical performance. The researchers hypothesize that (i) there is significant effects of MF on the basketball tactical performance; (ii) There is significant effects of brief mindfulness intervention on the basketball tactical performance after mental fatigue.

## 2. Method

### 2.1 Participants

The researchers recruited 54 male university students majoring in basketball training, who exhibited the following characteristics: age = 21.0 years (SD = 1.1), height = 183.4 cm (SD = 4.56), weight = 69.8 kg (SD = 6.6), and basketball training background of 53.6 months (SD = 9.5). All the participants were adults and signed the informed consent form of this study. The sample size was determined a priori using G*Power 3.1 software with effect size d = 0.5 (f = 0.25) [43].which required a minimum of 42 participants ($\alpha = 0.05$; [1-$\beta$] = 0.80; repeated measures ANOVA within-between interaction with two measurements; a correlation among repeated measures of 0.5) [44].

Additionally, to achieve the necessary statistical power in the context of a cluster randomized controlled trial (cRCT), the design effect (DE) was calculated using Equation 1: DE = 1 + $\rho$ (m—1) and Equation 2: ESS = DE * mk [M = number of participants in a cluster, k = number of clusters, mk = total number of participants in a cluster study, ESS = effective sample size, DE = design effect, $\rho$ = intra-cluster correlation coefficient (ICC)]. In this study,

an ICC value of 0.01 was employed [45]. Consequently, the recommended sample size was calculated as follows: ESS = [1 + 0.01 * (14–1)] * 14 * 3 = 47.46. Considering an expected dropout rate of 10%, the final sample size needed to be at least 52.74. As there were three groups in this study, 54 participants were enrolled.

All participants were recruited through social media platforms (e.g., WeChat) or emails from three universities (Wuhan Sports University, Hubei University, and Central China Normal University). The inclusion criteria were as follows: (i) male, (ii) age between 18 and 24, and (iii) no prior experience in mindfulness training. The exclusion criteria were: (i) individuals with medical prescriptions, (ii) color blindness, and (iii) participation in other studies or clinical trials. The 54 participants in three universities were randomly assigned to one of three groups: the control group (CG) (n = 18), the mental fatigue group (MFG) (n = 18), and the mental fatigue and mindfulness group (MF-MG) (n = 18).

## 2.2 Study design

The cluster randomised controlled trial (cRCT) design, with two intervention groups and one control group [46], was adopted to prevent contamination between the control and intervention groups in terms of space and time [47]. The Ethics Committee of University Putra Malaysia approved this study (Ethical approval reference number: JKEUPM-2021-889). The experiment of this study started from 11 May 2022 to 11 June 2022.

## 2.3 Measurements

The following instruments were used in this study:

1. **Visual Analog Scale (VAS):** Many studies have used the VAS to assess mental fatigue in sports [48, 49]. A 100 mm horizontal line was presented to the participants, with "100" indicating the highest level of mental fatigue and "0" indicating the lowest level. The VAS was also employed in this study to measure motivation [50].

2. **Attention Control Scale (ACS):** The ACS, developed by Derryberry & Reed in 2002, assesses participants' attentional focus and attentional shifting using 20 items rated on a 4-point Likert scale ranging from 1 (almost never) to 4 (always) [51]. In this study, the Chinese version of ACS [52] was used, which consists of 16 items following an analysis using Exploratory Structural Equation Modeling (ESEM). The scale demonstrated good internal consistency with α = 0.78 and a test-retest reliability of 0.72.

3. **Tactical Performance in Small Side Games (SSG):** SSGs are widely used to enhance technical and tactical skills in team sports [53]. The basketball SSG employed in this study was adapted from previous research and adhered to the following rules: (i) conducted on a half basketball court measuring 14 m × 15 m; (ii) three vs. three player formations; (iii) two sets of games, each lasting six minutes; (iv) no free throws (fouls resulted in normal sideline reposition); and (v) an 8-minute standardized warm-up session before the game. Tactical performance in this study was assessed based on various elements, including dribble penetration into the key area (DPKA), off-ball screen (OFFBS), ball reversal (BR), on-ball screen (ONBS), handoff (HO), and post entry (PE), following the guidelines of previous research [54–56].

4. **Five Facet Mindfulness Questionnaire (FFMQ):** The FFMQ, developed by Baer et al. (2006), was employed to evaluate participants' mindfulness state [57]. It assesses five facets, including non-judging of inner experience, describing, acting with awareness, observing, and non-reactivity to inner experience, through 39 items. Respondents rate their

experiences on a 5-point Likert scale ranging from 1 (not at all) to 5 (very much). This study utilized the Chinese version of the FFMQ [58], which exhibited good test-retest reliability (0.436–0.741, p < 0.01).

5. **Competitive State Anxiety Inventory-2 (CSAI-2):** The CSAI-2, developed by Martens et al. (1982), assesses participants' sport-specific anxiety, including somatic anxiety, cognitive anxiety, and self-confidence [59]. It comprises 27 items rated on a 4-point Likert scale, ranging from 1 (not at all) to 4 (very much). The Chinese version of CSAI-2 [60] was used in this study and exhibited good internal consistency reliability (0.68–0.72).

6. **Self-Control Scale (SCS):** Self-control was used as a covariate because of its relationship with sport performance and MF [25, 61, 62]. The SCS, developed by Tangney et al. (2004), measures self-control and consists of 19 items rated on a 5-point Likert scale, ranging from 1 (not at all) to 5 (very much) [63]. The Chinese version of SCS [64] was utilized in this study and demonstrated good internal consistency (α = 0.862) and test-retest reliability (0.850).

7. **Positive and Negative Affect Scale (PANAS):** The PANAS was employed to assess positive and negative moods [65]. Respondents indicated their current feelings on a 5-point Likert scale, ranging from 1 (not at all) to 5 (very much). This study utilized the Chinese version of PANAS with a Cronbach's α of 0.82 [66].

8. **Rating of Perception of Effort (RPE):** Borg's 6–20 scale was used to measure RPE [67]. Participants rated their perceived effort on a scale from 6 (no exertion at all) to 20 (maximal exertion). The Chinese version of the Borg scale, which exhibited a test-retest reliability of 0.94, was used in this study [67].

## 2.4 Experimental manipulation

1. **MF Inducement and Control Condition:** The incongruent Stroop task, lasting for 30 minutes, was created using the "E-Prime" software to induce mental fatigue [68–70]. This task requires continuous attention and forces the brain to manage conflicting stimuli, which increases cognitive load and depletes mental resources [31, 71]. Simultaneously, the congruent Stroop task, also lasting for the same duration, served as the control condition to avoid mental fatigue [25, 72]. During the task, words such as "green," "blue," "red," and "yellow" were displayed on the computer screen, with a 1.25-second rest interval. Participants were required to press differently colored buttons on the keyboard to respond. In the incongruent task, the different-colored buttons corresponded to different words, and the correct response was the button corresponding to the color of the word presented on the screen. In the congruent task, the buttons' color matched the words' color.

2. **Mindfulness Intervention and Control Condition:** A 30-minute mindfulness audio recording was utilized in the present study. This audio included (1) the centering exercise, (2) the mindfulness of the breath exercise, and (3) a body scan. The audio was recorded by a mindfulness instructor with over six years of experience in mindfulness interventions [73] (S1 Appendix). This intervention was based on the Mindfulness-Acceptance-Commitment (MAC) approach [74] and has been employed in numerous studies within the field of sports [75, 76].

The control condition for the mindfulness intervention involved reading a basketball magazine for the same duration. This activity aimed to induce a relaxed state [77].

## 2.5 Procedures

Two days before the experiment, participants underwent mindfulness training twice, conducted by a psychology teacher [25], and were introduced to the instruments. One day prior to the experiment, participants were instructed to ensure they had enough sleep (7–9 hours) and abstain from consuming caffeine and alcohol.

At the beginning of the experiment, participants took around six minutes to complete several questionnaires, including demographic information, visual analogue scale for motivation (VAS-MO), visual analogue scale for mental fatigue (VAS-MF), attention control scale (ACS), rating perception of effort (RPE), competitive state anxiety inventory-2 (CSAI-2), self-control scale (SCS), and five facet mindfulness questionnaire (FFMQ). Two instructors were arranged to help them finish these questionnaires as soon as possible in the necessary. Then, participants were guided by an 8-minute standardized warm-up characterized by several light cardio activities, such as jogging or skipping, to increase the heart rate gradually and prepare the body for movement, and some dynamic stretches targeting key muscle groups to enhance mobility and flexibility, including leg swings, arm circles, and lunges. The intensity of warm-up was controlled to avoid inducing MF. Afterwards, they were engaged in SSG. After the pre-test, participants in the MFG and MF-MG groups performed the incongruent Stroop task, while those in the CG completed the congruent Stroop task. Following this, participants filled out the positive and negative affect scale (PANAS) to assess whether the Stroop task had affected their moods. This measurement is deemed necessary because previous studies proved that overriding a well-learned behavior might have a negative effect on the emotional state [25, 78]. Subsequently, participants in the MF-MG group listened to a 30-minute mindfulness audio, while participants in the MFG and CG groups read basketball magazines. Finally, participants immediately took four minutes to complete FFMQ, VAS-MF, VAS-MO, ACS, RPE, and participated in SSG again.

## 2.6 Data analyses

All statistical analyses were performed using SPSS software (Version 26.0, SPSS Inc., Chicago, IL) with a significance level set at $P < 0.05$. The normality of all variables was assessed using the Shapiro–Wilk test. Levene's Test was used to check the homogeneity of variances. ANOVA was employed to assess homogeneity for dependent variables following a normal distribution, whereas Kruskal-Wallis H was used for variables not meeting the normal distribution. Generalized Estimating Equations (GEE) were used to analyze all hypotheses.

## 3. Results

Table 1 presents the mean ± SD of all variables. There were no significant differences among groups for age (F = 0.434, p = 0.650), height (F = 0.012, p = 0.988), weight (F = 1.021, p = 0.368), and training month (F = 0.617, p = 0.544). There were no significant differences among groups in terms of anxiety ($p \geq 0.05$) and self-control ($p \geq 0.05$). Similarly, no significant differences were observed in positive affective states (F = 2.015, p = 0.144) and negative affective states (F = 0.320, p = 0.728), indicating no unintended effects of the Stroop task on participants' mood.

GEE analysis was employed to assess whether there existed a difference in MF scores across groups and over time. As demonstrated in Table 2, the MF scores exhibited distinct patterns among the three groups over time due to the significant main effects of groups ($\chi2 = 65.661$, $p < 0.001$), time ($\chi2 = 98.470$, $p < 0.001$), and the interaction effect (group*time) ($\chi2 = 67.433$, $p < 0.001$) on MF scores. Fig 1(A) visually illustrates the successful induction of MF from the pre-test to the post-test and the effectiveness of mindfulness training as a recovery method.

**Table 1. Mean (± SD) score for variables.**

| Characteristics | CG | MFG | MF-MG | f | p-value |
|---|---|---|---|---|---|
| | Mean ± SD | Mean ± SD | Mean ± SD | | |
| Age (year) | 20.78 ± 1.00 | 21.11 ± 1.02 | 21.00 ± 1.24 | 0.434 | 0.650 |
| Height (cm) | 183.25 ± 3.67 | 183.49 ± 4.92 | 183.32 ± 5.19 | 0.012 | 0.988 |
| Weight (kg) | 77.94 ± 4.97 | 80.70 ± 7.83 | 80.67 ± 6.82 | 1.021 | 0.368 |
| Training (month) | 53.33 ± 10.15 | 55.50 ± 8.81 | 52.00 ± 9.62 | 0.617 | 0.544 |
| Anxiety (total) | 52.56 ± 7.579 | 51.89 ± 5.614 | 52.72 ± 6.341 | 0.081 | 0.922 |
| Self-control | 53.67 ± 7.121 | 53.83 ± 7.294 | 53.11 ± 7.638 | 0.048 | 0.954 |
| Positive Mood | 22.50 ± 3.204 | 20.39 ± 3.109 | 21.17 ± 3.258 | 2.015 | 0.144 |
| Negative Mood | 15.06 ± 2.879 | 14.5 ± 2.572 | 15.17 ± 2.572 | 0.320 | 0.728 |

*Note.* CG = control group; MFG = mental fatigue group; MF-MG = mental fatigue and mindfulness group

Table 2 demonstrates distinct patterns in attention focus and attention shift scores among the three groups over time, as evidenced by the significant interaction effects (Group*Time) on attention focus ($\chi2$ = 175.790, p < 0.001) and attention shift ($\chi2$ = 200.963, p < 0.001), respectively. Fig 1(B) and 1(C) visually depict how MF influenced attention focus and shift, with mindfulness training effectively aiding participants in recovering from MF by preserving their attention focus and shift.

According to the results of GEE on total tactics (Table 2), the main effect of group ($\chi^2$ = 2.661, p = 0.264) and time ($\chi^2$ = 1.411, p = 0.235) on the number of tactics was not statistically significant. Based on these results, the interaction effect (Group*Time) on the number of tactics was insignificant ($\chi^2$ = 2.446, p = 0.294), indicating that the three groups had no significantly different number of tactics across time.

Although the total tactics did not show the differences among groups and time, whether there was any difference in sub-variables of total tactics was unclear. The GEE was applied to determine if there was a difference in specific tactical performance among groups. Table 2 reveals distinct patterns in ball reversal (BR), dribble penetration into the key area (DPKA), on-ball screen (ONBS), and off-ball screen (OFFBS) among the three groups over time. Fig 1(D) to 1(G) illustrate the mean number of BR, DPKA, ONBS, and OFFBS across time. Specifically, the number of BR significantly decreased in the Mental Fatigue Group (MFG), with minimal changes in the Control Group (CG) and Mental Fatigue-Mindfulness Group (MF-MG) from the pre-test to the post-test. Conversely, the number of DPKA substantially increased in MFG, while showing slight changes in MF-MG from pre-test to post-test. The number of ONBS significantly decreased in MFG, with only minor changes in CG and MF-MG from pre-test to post-test. Finally, there were minimal changes in CG, MF-MG, and MF-MG in the number of OFFBS from pre-test to post-test. Notably, in the post-test, significant differences were observed between CG and MFG, as well as CG and MF-MG, for BR, DPKA, ONBS, and OFFBS. These findings suggest that mental fatigue (MF) had a notable impact on the performance of BR, DPKA, ONBS, and OFFBS, while mindfulness training contributed to their recovery.

Table 2 highlights the significant interaction effect (Group*Time) on mindfulness state scores ($\chi2$ = 1150.906, p < 0.001), indicating notable differences in scores among the three groups across time. Fig 1(H) vividly illustrates these differences, particularly between the Mental Fatigue Group (MFG) and the Mental Fatigue-Mindfulness Group (MF-MG). It is evident that mindfulness state scores significantly varied between MFG and MF-MG over time. These findings underscore the adverse effects of mental fatigue (MF) on mindfulness state scores and the protective influence of mindfulness training in mitigating these adverse effects.

**Table 2. Descriptive statistics (Mean ± SD) and GEE results on variables.**

| Variable | Group | Pre | Post | Effects | Wald Chi-Square | P value |
|---|---|---|---|---|---|---|
| | | Mean (SD) | Mean (SD) | | | |
| MF | CG | 2.41 ± 0.32 | 2.57 ± 0.42 | Group | 65.661* | <0.001 |
| | MFG | 2.56 ± 0.39 | 3.62 ± 0.24 | Time | 98.470* | <0.001 |
| | MF-MG | 2.37 ± 0.36 | 2.50 ± 0.31 | Interaction | 67.433* | <0.001 |
| Attention focus | CG | 24.17 ± 2.77 | 23.94 ± 2.86 | Group | 6.255* | <0.001 |
| | MFG | 23.78± 2.94 | 20.17 ± 2.48 | Time | 243.824* | 0.041 |
| | MF-MG | 24.17± 2.52 | 23.22± 2.20 | Interaction | 175.437* | <0.001 |
| Attention shift | CG | 24.17 ± 2.75 | 23.89 ± 2.74 | Group | 5.208 | 0.074 |
| | MFG | 23.67 ± 2.71 | 20.56 ± 2.79 | Time | 111.692* | <0.001 |
| | MF-MG | 23.83 ± 2.38 | 23.56 ± 2.41 | Interaction | 200.963* | <0.001 |
| Tactical performance (total) | CG | 23.44 ± 6.13 | 22.61 ± 3.61 | Group | 2.661 | 0.264 |
| | MFG | 21.72 ± 3.67 | 20.61 ± 3.33 | Time | 1.411 | 0.235 |
| | MF-MG | 22.39 ± 3.64 | 22.67 ± 2.71 | Interaction | 2.446 | 0.294 |
| BR | CG | 2.89 ± 1.10 | 2.61 ± 0.59 | Group | 7.804* | 0.020 |
| | MFG | 2.44 ± 1.06 | 1.89 ± 0.74 | Time | 6.676* | 0.010 |
| | MF-MG | 3.00 ± 1.54 | 2.67 ± 0.67 | Interaction | 0.889 | 0.641 |
| DPKA | CG | 9.50 ± 2.61 | 9.11 ± 1.45 | Group | 6.636* | 0.036 |
| | MFG | 9.50 ± 1.92 | 11.06 ± 2.15 | Time | 1.418 | 0.234 |
| | MF-MG | 8.89 ± 1.79 | 8.67 ± 1.73 | Interaction | 15.803* | <0.001 |
| PE | CG | 2.06 ± 0.97 | 1.61 ± 0.89 | Group | 4.902 | 0.086 |
| | MFG | 1.44 ± 0.76 | 1.72 ± 0.73 | Time | 0.635 | 0.426 |
| | MF-MG | 2.06 ± 0.78 | 1.83 ± 0.90 | Interaction | 3.363 | 0.186 |
| ONBS | CG | 5.06 ±2.6 | 5.61 ± 2.65 | Group | 2.384 | 0.304 |
| | MFG | 5.00 ± 2.03 | 3.50 ± 1.54 | Time | 0.112 | 0.737 |
| | MF-MG | 4.33± 1.41 | 5.11 ± 1.33 | Interaction | 42.109* | <0.001 |
| OFFBS | CG | 2.50 ± 1.43 | 2.44 ± 1.12 | Group | 34.022* | <0.001 |
| | MFG | 1.83 ± 0.76 | 1.44 ± 0.76 | Time | 0.104 | 0.747 |
| | MF-MG | 2.50 ± 0.69 | 2.78 ± 0.92 | Interaction | 3.176 | 0.204 |
| HO | CG | 1.44 ± 0.83 | 1.22 ± 0.71 | Group | 5.253 | 0.072 |
| | MFG | 1.50 ± 0.69 | 1.00 ± 0.67 | Time | 2.933 | 0.087 |
| | MF-MG | 1.61 ± 0.59 | 1.61 ± 0.67 | Interaction | 2.226 | 0.329 |
| Mindfulness state | CG | 76.50 ± 7.48 | 76.89 ± 7.88 | Group | 12.712* | 0.002 |
| | MFG | 77.61 ± 6.98 | 69.22 ± 7.09 | Time | 1.703 | 0.192 |
| | MF-MG | 78.28 ± 7.77 | 85.61 ± 7.12 | Interaction | 1150.906* | <0.001 |
| RPE | CG | 7.72 ± 1.32 | 11.89 ± 1.52 | Group | 20.877* | <0.001 |
| | MFG | 8.28 ± 1.52 | 15.11 ± 1.33 | Time | 1416.278* | <0.001 |
| | MF-MG | 7.67 ± 1.37 | 14.83 ± 1.07 | Interaction | 72.822* | <0.001 |
| Motivation | CG | 6.02 ± 0.09 | 5.94 ± 0.10 | Group | 0.564 | 0.754 |
| | MFG | 6.13 ± 0.10 | 5.96 ± 0.10 | Time | 11.069* | 0.001 |
| | MF-MG | 5.98 ± 0.10 | 5.90 ± 0.11 | Interaction | 1.490 | 0.475 |

*Note*. CG = control group; MFG = mental fatigue group; MF-MG = mental fatigue and mindfulness group; MF = mental fatigue; RPE = rating perception of effort; DPKA = dribble penetration into the key area; OFFBS = off-ball screen; BR = ball reversal; ONBS = on-ball screen; HO = handoff; PE = post entry

Table 2 illustrates that the interaction effect (Group*Time) was significant ($\chi 2$ = 72.822, p < 0.001) on the scores of RPE, but not on motivation ($\chi^2$ = 1.490, p = 0.475). Fig 1(I) shows that the RPE scores were significantly different between times across all groups.

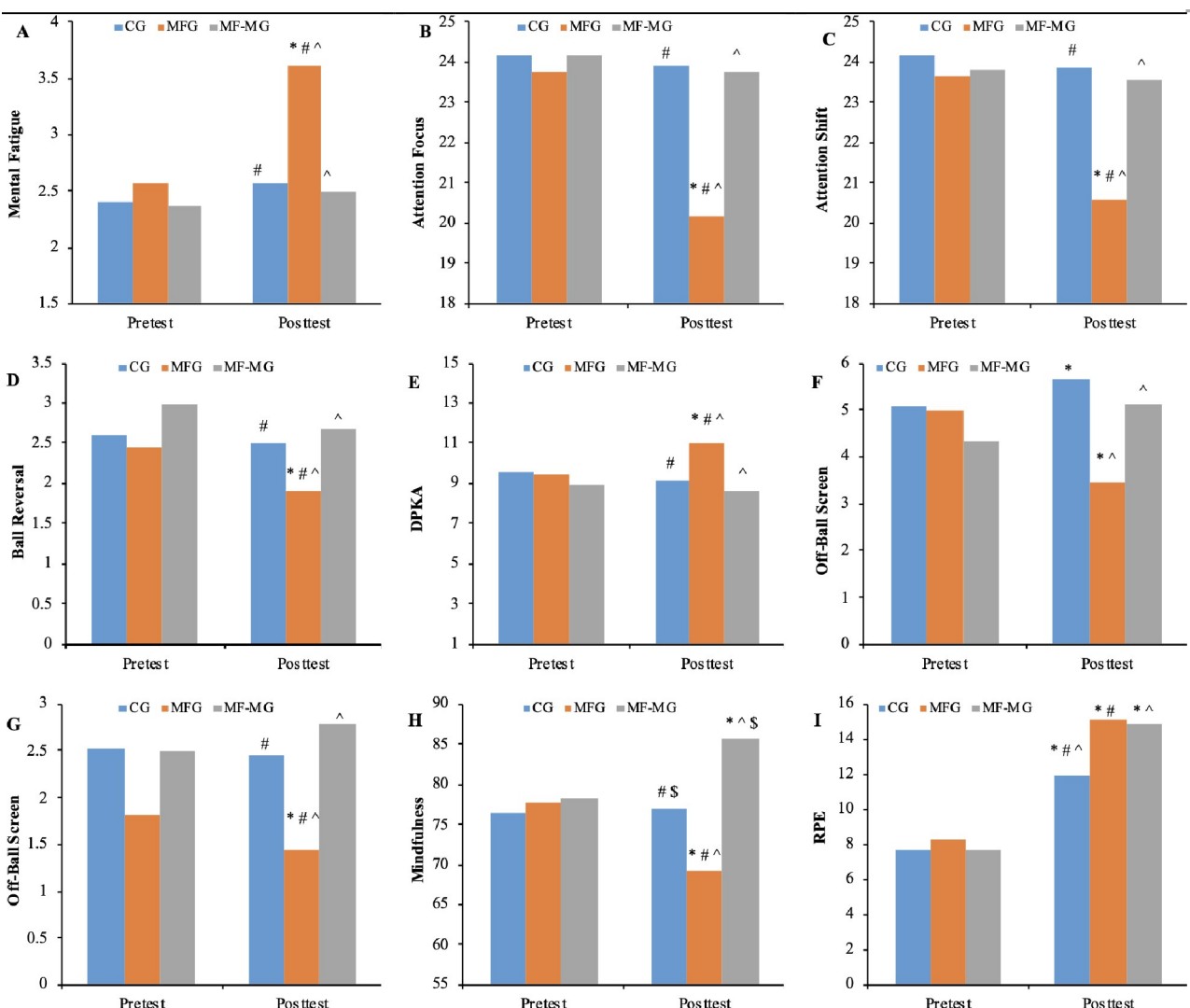

**Fig 1.** Interaction effect for (A) mental fatigue; (B) attention focus; (C) attention shift; (D) ball reversal; (E) dribble penetration into the key area; (F) on-ball screen; (G) off-ball screen; (H) mindfulness; (I) rating perception of effort. CG, control group; MFG, mental fatigue group; MF-MG, mental fatigue and mindfulness group; *, The values between pre-test and post-test were significant; #, The values between MFG and CG were significant; ^, The values between MFG and MF-MG was significant; $ = The values between CG and MF-MG were significant.

## 4. Discussion

The current study found no significant difference in total tactics across groups and time. However, when examining sub-variables of tactics, notable differences emerged. Specifically, there was a significant discrepancy in ball reversal (BR), dribble penetration into the key area (DPKA), and off-ball screen (OFFBS) between the Control Group (CG) and the Mental Fatigue Group (MFG) in the post-test. These results suggest that MF adversely affected players' tactical performance. Furthermore, there were significant differences in BR, DPKA, on-ball screen (ONBS), and OFFBS between MFG and the Mental Fatigue-Mindfulness Group (MF-MG) in the post-test, indicating that mindfulness training played a restorative role in enhancing tactical performance.

Notably, this study represents a pioneering effort in investigating the impact of MF on basketball tactical performance in actual gameplay. Although the total number of tactics did not

significantly differ between CG and MFG, it's crucial to emphasize that the total number of tactics alone does not determine the quality of tactical performance. Previous research has shown that losing teams often exhibit lower BR and post entry (PE) counts along with higher DPKA and OFFBS counts, possibly due to their reliance on defensive zone strategies [54]. However, this study exclusively employed man-to-man defensive strategies, potentially explaining the divergent results compared to previous research [79].

Tactical skills in basketball are closely linked to decision-making abilities. Prior studies have demonstrated that MF can impair decision-making speed in basketball players [27]. Decision-making speed likely contributed to the differences observed between CG and MFG in this study. Similarly, MF has been shown to compromise decision-making accuracy and speed in soccer players [80]. A recent study proved that players could focus more on relevant information in a decision-making task with the improvement of directed attention [81]. However, the prior research primarily examined the effects of MF on decision-making in controlled tasks [80, 81], but the current study delved into its impact on actual game performance. It is plausible that MF hinders decision-making during games, possibly affecting players' perception and the effectiveness of actions such as dribbling, passing, and movement. For instance, dribbling may not create the necessary space in the opponent's defence, which could challenge scoring opportunities. In this context, players with MF might pay increased attention to the opponent's body positioning, potentially impeding the decision-making process. Environmental factors, including audience and opponent pressure, could further exacerbate these challenges, potentially explaining the differences between CG and MFG in BR, DPKA, PE, and screen. Therefore, it can be inferred that MF adversely affects basketball tactical performance.

These impairments in decision-making associated with MF may be linked to specific brain regions, such as the left frontoparietal region, which is critical in the context of decision-making [82]. Visual search, which integrates and processes somatosensory and sensorimotor information, is highly supported by various brain region networks [82]. A study has even suggested that MF can alter visual behavior and hinder decision-making in basketball players [83]. The present study has also corroborated that MF can disrupt players' attention. Consequently, players experiencing MF may struggle to track the positions of other players and maintain fixations on relevant cues [84].

The current study marks a novel exploration into the effects of mindfulness training on the recovery of tactical performance in actual basketball games, as initially researched by Shudian et al. (2022) [23]. The intervention significantly improved attention levels impaired by MF, thereby enhancing decision-making and tactical performance in basketball. This aligns with the Mindfulness and De-Automatization model proposed by Kabat-Zinn et al. (1985) and Teasdale et al. (1995), as well as with findings from other studies [85, 86]. Specifically, mindfulness is shown to induce four key mental processes: preventing thought distortion and suppression, facilitating metacognitive insight, enhancing cognitive control, and reducing automatic inference processing. Kang et al. (2012) highlighted that these processes promote health outcomes and adaptive self-regulation through de-automatizing functions [87].Furthermore, numerous studies, including those by Alfonso et al. (2011) and Liu et al. (2018), have demonstrated mindfulness's influence on factors like attention, compassion, and cognitive control within decision-making processes [88, 89]. In a related vein, Zhu et al. (2020) and Coimbra et al. (2021) employed an assessment tool to measure the MF score, noting a significant increase post-mindfulness intervention [34, 35]. Axelsen et al. (2020) illustrated that four weeks of mindfulness training could reduce scores on the Sustained Attention to Response Task, a tool for assessing MF [36]. In the realm of sports, studies by Shaabani et al. (2020) and Wang et al. (2017) revealed that several weeks of mindfulness intervention post-MF could respectively recover basketball free-throw and handgrip performance [25, 90].

According to the results, coaches should recognize that MF can diminish tactical execution and be prepared to identify and manage it. Prior to games, incorporating brief mindfulness sessions can help players maintain mental clarity and optimal focus, potentially preventing tactical errors stemming from MF. On the other hand, trainers could incorporate mindfulness training into the regular training regimen of basketball players. For instance, combining tactical drills with mindfulness exercises could lead to more focused and effective tactical practice.

## 5. Limitations

While this study has successfully demonstrated the effectiveness of mindfulness in mitigating mental fatigue (MF) in a real basketball game setting, its use of the MF-inducing Stroop task poses a limitation, as it is not directly related to game-specific tasks. Future research should aim to identify and utilize more game-relevant tasks to induce MF. Moreover, this study focused solely on male basketball players. Future research could benefit from exploring the differential effects of mindfulness interventions across genders. Lastly, testing every basketball tactic in a single study is impractical. Thus, there is a need for more comprehensive studies that focus on various aspects of mindfulness and its impact on basketball performance.

## 6. Conclusion

This study is the first to investigate the effects of a brief mindfulness intervention on basketball players' tactical performance in real games. It was found that certain tactical elements including ball reversal, dribble penetration into the key area, on-ball screen and off-ball screen were negatively impacted by MF, and that a brief mindfulness intervention could facilitate their recovery. Coaches and trainers should consider incorporating mindfulness therapy into their basketball training routines. Future research should aim to identify additional methods to enhance players' psychological states, thereby improving their on-site performance in competitions. Researchers should focus on exploring the impact of mindfulness interventions on a broader range of basketball tactics beyond the present study, like fast breaks, zone defense, transition defense, and set plays, leading to a comprehensive tactical analysis.

## Supporting information

**S1 Appendix. Mindfulness intervention.**
(PDF)

## Acknowledgments

The authors would like to thank Qian Shaowen and Wu Jianxi for their assistance with providing the basketball court.

## Author Contributions

**Conceptualization:** Shudian Cao.

**Data curation:** Shudian Cao.

**Formal analysis:** Shudian Cao.

**Funding acquisition:** Shudian Cao, Jia Liu.

**Investigation:** Jia Liu, He Sun, Xiaopeng Wang.

**Methodology:** Jia Liu, He Sun, Xiaopeng Wang.

**Project administration:** He Sun, Xiaopeng Wang.

**Resources:** Xiaopeng Wang.

**Software:** Xiaopeng Wang.

**Supervision:** Soh Kim Geok, Xiaopeng Wang.

**Validation:** Soh Kim Geok.

**Visualization:** Soh Kim Geok.

**Writing – original draft:** Shudian Cao.

**Writing – review & editing:** Shudian Cao.

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
