## [Decision Letter · Decision Letter 0]

8 May 2024

PONE-D-24-08570Effects of Brief Mindfulness Intervention on Mental Fatigue and Recovery in Basketball Tactical PerformancePLOS ONE

Dear Dr. Cao,

Thank you for submitting your manuscript to PLOS ONE. After careful consideration, we feel that it has merit but does not fully meet PLOS ONE’s publication criteria as it currently stands. Therefore, we invite you to submit a revised version of the manuscript that addresses the points raised during the review process.

We look forward to receiving your revised manuscript.

Kind regards,

Vanessa Carels

Staff Editor

PLOS ONE

Journal Requirements:

Reviewers' comments:

Reviewer's Responses to Questions

**Comments to the Author**

1. Is the manuscript technically sound, and do the data support the conclusions?

Reviewer #1: Yes

Reviewer #2: Yes

2. Has the statistical analysis been performed appropriately and rigorously? 

Reviewer #1: Yes

Reviewer #2: Yes

3. Have the authors made all data underlying the findings in their manuscript fully available?

Reviewer #1: Yes

Reviewer #2: Yes

4. Is the manuscript presented in an intelligible fashion and written in standard English?

Reviewer #1: Yes

Reviewer #2: Yes

5. Review Comments to the Author

**Reviewer #1: **Thank you for the opportunity to review the article entitled - Effects of Brief Mindfulness Intervention on Mental Fatigue and Recovery in Basketball Tactical Performance. The article is interesting and to increase the scientific level I believe that the following recommendations can be taken into account.

Introduction

- The new aspects of the study compared to previous studies should be highlighted in more detail.

Material and method

- It is well structured and detailed.

Results

- They are well interpreted, and the tables are easy to view.

Discussions

- Lines 344-346 need references

- To explain the practical implications of the most relevant results of the study on the training of basketball players.

Conclusions.

Future research directions to be added.

**Reviewer #2**: Dear Authors,

First of all, I would like to express my appreciation for the profound work you have presented in your manuscript entitled "Effects of Brief Mindfulness Intervention on Mental Fatigue and Recovery in Basketball Tactical Performance." The subject is highly pertinent and your approach to exploring the effects of mindfulness interventions on athletic performance is both innovative and timely. Your study brings valuable insights into the fields of sports psychology and performance enhancement.

In the process of review, I have several observations that could further solidify the introduction of your work and ensure it adequately sets the stage for your findings. Here are some specific points for your consideration:

Introduction:

Lack of references to original research: the introduction relies quite heavily on review articles rather than direct empirical studies, which could diminish the empirical foundation of your research. I recommend including specific original research studies for each mentioned effect of mental fatigue.

Point of insertion for the discussion on reaction time: currently, there is an absence of discussion on the impact of mental fatigue on reaction times, an essential factor in sports like basketball. It would be beneficial to add a section that discusses previous findings on reaction times under mental fatigue.

Methodology:

Clarity and justification of the Stroop Task: the justification for using the Stroop Task to induce mental fatigue lacks detailed explanation. I advise strengthening this section by citing studies that have utilized similar methodologies.

Data collection and protocol: the manuscript would benefit from more detailed descriptions of the data collection methods and experimental protocols. Enhancing the description of the experimental setup and data collection procedures would be highly advantageous.

Results:

Presentation and clarity of data: the presentation of the results could be clearer. I advise reorganizing this section to include more detailed tables and clearer figures.

Discussion and Conclusion:

Alignment of the discussion with the results: it is crucial to ensure that the discussion remains closely aligned with the empirical findings. I suggest revising this section to avoid any overgeneralization not directly supported by the results.

Consistency and concluding remarks: the conclusion might not effectively encapsulate the research findings. I recommend providing a more concise summary of the findings in the conclusion.

I trust these observations will assist in enhancing the clarity, depth, and scientific integrity of your manuscript. The points provided are intended as suggestions to consider. My overall assessment leads me to recommend a minor revision; however, I would be gratified to see a careful reconsideration of these points in your revision process. I look forward to your thoughtful engagement with these suggestions, which I believe will contribute significantly to the literature on mindfulness and athletic performance.

6. PLOS authors have the option to publish the peer review history of their article (what does this mean?). If published, this will include your full peer review and any attached files.

Reviewer #1: No

Reviewer #2: No

---

## [Author Response · Author response to Decision Letter 0]

13 May 2024

Dear editor Vanessa Carels,

Thanks for your patience with our paper. We revised the paper according to your and the reviewers’ comments, And a revised manuscript with the correction sections yellow marked was attached as the supplemental material and for easy check/editing purpose. We hope you are satisfied with our revisions and our revised version will be satisfactory for publication in PLOS ONE. And please do not hesitate to contact us if you have any new suggestions. Thanks again.

Best regards

Sincerely yours

Dear reviewer 1,

Thank you very much for your attention, evaluation and comments on our paper. We revised the manuscript in accordance with your kind and detailed comments, and carefully proof-read the manuscript to minimize typographical, grammatical, and bibliographical errors. We made many revisions according to your comments. A document answering every question from you was summarized and enclosed. And a revised manuscript with the correction sections yellow marked was attached as the supplemental material and for easy check/editing purpose.

Introduction

1. The new aspects of the study compared to previous studies should be highlighted in more detail.

Line 73-79. Line 90-94. Line 97-103. According to the advice, more detailed introduction was added. 

Material and method

2. It is well structured and detailed.

Thanks.

Results

3. They are well interpreted, and the tables are easy to view.

Thanks.

Discussions

4. Lines 344-346 need references

Line 366. The reference was added.

5. To explain the practical implications of the most relevant results of the study on the training of basketball players.

Line 418-424. According to this suggestion, some practical implications were added. 

Conclusions.

6. Future research directions to be added.

Line 441-446. The future directions were added here.

Finally, we are very grateful to the reviewer for the detailed and kind suggestions. We are deeply inspired. We hope that the reviewer will be satisfied with our revisions and our revised version will be satisfactory for publication in PLOS ONE. Please do not hesitate to contact us if you have any new suggestions. Great thanks to you. 

Best regards

Sincerely yours

Dear reviewer 2, 

Thank you very much for your attention, evaluation and comments on our paper. We made many revisions in accordance with your kind and detailed comments, and carefully proof-read the manuscript to minimize typographical, grammatical, and bibliographical errors. A document answering every question from you was summarized and enclosed. And a revised manuscript with the correction sections red marked was attached as the supplemental material and for easy check/editing purpose.

Introduction:

1. Lack of references to original research: the introduction relies quite heavily on review articles rather than direct empirical studies, which could diminish the empirical foundation of your research. I recommend including specific original research studies for each mentioned effect of mental fatigue.

Line 51-55. Line 73-79. Line 90-94. Line 97-103. According to this suggestion, some empirical references were added in this part.

2. Point of insertion for the discussion on reaction time: currently, there is an absence of discussion on the impact of mental fatigue on reaction times, an essential factor in sports like basketball. It would be beneficial to add a section that discusses previous findings on reaction times under mental fatigue.

Line 90-94. The effect of mental fatigue on reaction time was explained here.

Methodology:

3. Clarity and justification of the Stroop Task: the justification for using the Stroop Task to induce mental fatigue lacks detailed explanation. I advise strengthening this section by citing studies that have utilized similar methodologies.

Line 223-227. More description of Stroop task was introduced, and many related references were also added here.

4. Data collection and protocol: the manuscript would benefit from more detailed descriptions of the data collection methods and experimental protocols. Enhancing the description of the experimental setup and data collection procedures would be highly advantageous.

Line 259-264. More details of the experiment were added here. And the information about the experimental protocols were added in the Appendix. Thank you very much.

Results:

5. Presentation and clarity of data: the presentation of the results could be clearer. I advise reorganizing this section to include more detailed tables and clearer figures.

All the related tables and figures were provided in this manuscript. Because we have many tables and figures, we make them together for the readers to see. And we have bolded all the symbols in the figures for readers to see it clearly. If you still think we lack some tables or figures, please tell us without any hesitate, we will improve it. Thank you very much.

Discussion and Conclusion:

6. Alignment of the discussion with the results: it is crucial to ensure that the discussion remains closely aligned with the empirical findings. I suggest revising this section to avoid any overgeneralization not directly supported by the results.

According to the advice, we deleted some unrelated sentences and also added some related sentences according to your and another reviewer’s advice (e.g. Line 418-424). 

7. Consistency and concluding remarks: the conclusion might not effectively encapsulate the research findings. I recommend providing a more concise summary of the findings in the conclusion.

Line 441-446. According to the advice, we revised this part.

Finally, we are very grateful to the reviewer for the detailed and kind suggestions. We are deeply inspired. We hope that the reviewer will be satisfied with our revisions and our revised version will be satisfactory for publication in PLOS ONE. And please do not hesitate to contact us if you have any new suggestion. Great thanks to you. 

Best regards

Sincerely yours

---

## [Decision Letter · Decision Letter 1]

25 Jun 2024

Effects of Brief Mindfulness Intervention on Mental Fatigue and Recovery in Basketball Tactical Performance

PONE-D-24-08570R1

Dear Dr. Cao,

We’re pleased to inform you that your manuscript has been judged scientifically suitable for publication and will be formally accepted for publication once it meets all outstanding technical requirements.

Kind regards,

Hesam Ramezanzade, Ph.D

Academic Editor

PLOS ONE

Additional Editor Comments (optional):

Reviewers' comments:

Reviewer's Responses to Questions

**Comments to the Author**

1. If the authors have adequately addressed your comments raised in a previous round of review and you feel that this manuscript is now acceptable for publication, you may indicate that here to bypass the “Comments to the Author” section, enter your conflict of interest statement in the “Confidential to Editor” section, and submit your "Accept" recommendation.

Reviewer #1: All comments have been addressed

Reviewer #2: All comments have been addressed

Reviewer #3: All comments have been addressed

2. Is the manuscript technically sound, and do the data support the conclusions?

Reviewer #1: Yes

Reviewer #2: Yes

Reviewer #3: Yes

3. Has the statistical analysis been performed appropriately and rigorously? 

Reviewer #1: Yes

Reviewer #2: N/A

Reviewer #3: Yes

4. Have the authors made all data underlying the findings in their manuscript fully available?

Reviewer #1: Yes

Reviewer #2: Yes

Reviewer #3: Yes

5. Is the manuscript presented in an intelligible fashion and written in standard English?

Reviewer #1: Yes

Reviewer #2: Yes

Reviewer #3: Yes

6. Review Comments to the Author

Reviewer #1: The authors improved the article according with the recommendations. The article in its current form meets the journal's requirements for publication. No further comments.

Reviewer #2: I have carefully reviewed the changes and responses you have provided, and I believe that in its current form, the work can make a significant scientific contribution.

Kind regards

Reviewer #3: There are no comments for the authors. The manuscript has been well revised and its current form can be published in the journal.

7. PLOS authors have the option to publish the peer review history of their article (what does this mean?). If published, this will include your full peer review and any attached files.

Reviewer #1: **Yes: **Badau Dana

Reviewer #2: **Yes: **Gian Mario Migliaccio

Reviewer #3: No

---

## [Editor Report · Acceptance letter]

18 Oct 2024

PONE-D-24-08570R1 

PLOS ONE

Dear Dr. Cao, 

I'm pleased to inform you that your manuscript has been deemed suitable for publication in PLOS ONE. Congratulations! Your manuscript is now being handed over to our production team.

Kind regards, 

on behalf of

Dr. Hesam Ramezanzade 

Academic Editor

PLOS ONE